# Low Velocity Impact Monitoring of Composite Tubes Based on FBG Sensors

**DOI:** 10.3390/s24041279

**Published:** 2024-02-17

**Authors:** Shengsheng Huan, Linjiao Lu, Tao Shen, Jianke Du

**Affiliations:** Smart Materials and Advanced Structure Laboratory, School of Mechanical Engineering and Mechanics, Ningbo University, Ningbo 315211, China; 2111081104@nbu.edu.cn (S.H.); 2111081018@nbu.edu.cn (L.L.)

**Keywords:** composite material, fiber Bragg grating, neural network, outlier, impact localization

## Abstract

Carbon fiber reinforced composites (CFRP) are susceptible to hidden damage from low velocity external impacts during their service life. To ensure the proper monitoring of the state of the composites, it is crucial to predict the location of an impact event. In this paper, fiber Bragg grating (FBG) sensors are affixed to the surface of a carbon fiber composite tube, and an optical sensing interrogator is used to capture the central wavelength shift of the FBG sensors due to low-velocity impacts. A discrete wavelet transform is used for noise reduction in the response signals. Then, the differences in the captured response signals of the FBG sensors at different locations of the impact were analyzed. Moreover, two methods were implemented to predict the location of low-velocity impacts, according to the differences in the captured response signals. The BP neural network-based method utilized three data sets to train the neural network, resulting in an average localization error of 20.68 mm. In contrast, the method based on error outliers selected a specific data set as the reference dataset, achieving an average localization error of 13.98 mm. The comparison of the predicted results shows that the latter approach has a higher predictive accuracy and does not require a significant amount of data.

## 1. Introduction

Carbon fiber reinforced composites have gained widespread adoption in various industries, including vehicle transportation [1,2], military equipment [3,4], and aerospace [5,6], owing to their inherent benefits such as a high, specific strength and stiffness. However, the utilization of carbon fiber composites inevitably exposes them to low-velocity impact loads during operational usage, which can result in concealed internal damage and a notable degradation in the mechanical properties of the structure [7,8,9]. It has been observed that composites are particularly vulnerable to lateral impact loads, and there is evidence to suggest that repeated impacts can induce damage, even in the absence of visible signs, following a single impact event [10,11]. Consequently, it is of paramount importance to proactively monitor impact events during the service life of composite structures to mitigate the risk of impact-induced damage [12].

In recent years, there has been a growing interest in the application of sensors for structural health monitoring and early prediction. Numerous scholars have conducted extensive research on the localization of low-velocity impact damage in composite structures, proposing several meaningful localization methods, including the use of strain gauges [13], accelerometers [14], piezoelectric ceramic patches [15], and piezoelectric film sensors [16]. With the continuous development of fiber optic sensing technology, various key technologies have gradually matured and been applied in practice [17,18,19]. Fiber optic sensing technology is widely regarded as a very promising technology. Among them, FBG sensors have attracted more attention in health monitoring due to their small size, high temperature resistance, electromagnetic interference resistance, and ease of implementation [20,21,22,23].

Previous studies have demonstrated the role of FBG sensors in structural health monitoring. Impact localization can be achieved by measuring the flight time of ultrasonic Lamb waves [24,25], using reference database-based impact localization algorithms [26,27], or utilizing strain amplitude [28] and other techniques. Xianglong Wen et al. [29] optimized the structure of the FBG sensing network, extracted feature vectors from impact signals, and achieved an average positioning error of 2.1 cm using a backpropagation (BP) neural network model. Pratik Shrestha et al. [30] proposed and implemented an algorithm based on error outliers, achieving an average prediction error of 10.7 mm for impact points on composite panels.

FBG sensors have received significant attention in the research of impact localization in plate-like structures. However, composites of tubular structures also have a wide range of applications in engineering [31,32,33], but there are fewer studies on the monitoring of low-velocity impacts of tubular composites. The effectiveness of methods such as neural networks in the health monitoring of tubular structures is yet to be verified. In this paper, four FBG sensors are bonded to the surface of carbon fiber composite pipe structures. The response signals detected are subjected to noise reduction processing. The differences in response signals captured by the FBG sensors at different positions when subjected to the impacts are analyzed. Subsequently, the low-velocity impact locations are predicted using both a BP neural network-based approach and an approach based on error outliers.

The content of this paper is divided into six sections. Section 2 provides an overview of the basic principles of FBG sensors and introduces the use of discrete wavelet transforms for noise removal. Section 3 describes the detailed experimental procedures and results. In Section 4, the BP neural network method is employed to predict the positions of low-velocity impacts. Section 5 outlines the algorithm based on error outliers and utilizes this algorithm to predict the positions of low-velocity impacts. Finally, the conclusion is presented in Section 6.

## 2. Theory of Impact Monitoring

### 2.1. Fundamentals of FBG

The sensing mechanism of the FBG sensor is as follows: When light passes through the FBG sensor, the light that fulfills the Bragg conditions experiences a strong reflection and is influenced accordingly, whereas the light that fails to meet these conditions undergoes minimal changes and transmits out of the fiber. The wavelength of the reflected light is referred to as the center wavelength. According to the coupled mode theory, the center wavelength of the FBG sensor can be expressed as [34]:(1)$$\lambda_{B}=2n_{eff}\Lambda$$
where the $\lambda_{B}$ in Equation (1) represents the center wavelength of the FBG sensor, $n_{eff}$ denotes the effective refractive index of the fiber core, and Λ signifies the cycle length of the grating. Any alterations in the external stress strain or temperature can induce a displacement in the center wavelength of the FBG sensor. When solely considering the influence of strain, the correlation between the wavelength shift at the center of the FBG sensor and the strain can be expressed as:(2)$$∆\lambda_{B}=\lambda_{B}(1-P_{e})\varepsilon$$
where the $P_{e}$ in Equation (2) represents the valid elastic-optic constants of the grating, which remains constant for gratings made of the same material. Equation (2) demonstrates a linear relationship between the center wavelength of the FBG sensor and the strain. When the composite material experiences micro-deformation due to impact, it induces micro-strain in the FBG sensor, resulting in a change in the center wavelength. Therefore, the FBG sensor can be effectively employed for the monitoring of low-velocity impact events.

### 2.2. Discrete Wavelet Transform Denoising

During practical usage, the response signal obtained from the FBG sensor through the optical sensing interrogator often contains a significant amount of noise. Failure to eliminate this noise can severely impact signal processing and analysis. Common signal denoising methods include Moving Average (MA), Moving Difference (MD), Frequency Domain Dynamic Average (FDDA), Activation Function Dynamic Average (ADFA), and Discrete Wavelet Transform (DWT), among others. In this study, we have chosen to use DWT for denoising the FBG signals. The DWT exhibits excellent time-frequency localization properties and is capable of effectively preserving the peaks and abrupt changes within the original signal. Therefore, the DWT is a suitable choice for denoising the signal. The denoising process using the DWT can be divided into three steps: firstly, obtaining the smoothing coefficients and wavelet coefficients through multistage wavelet decomposition; secondly, analyzing the wavelets and selecting an appropriate thresholding technique; and finally, applying the thresholding to the wavelet coefficients and reconstructing the denoised signal.

The thresholding process primarily involves two aspects: the determination of the threshold value and the selection of the threshold function. The threshold value serves as a criterion for distinguishing noise, and its magnitude directly affects the quality of denoising. In this study, the Heursure threshold is adopted, which combines elements from both the rigsure threshold and the sqtwolog threshold. The specific form of the Heursure threshold begins by comparing two variables, denoted as β and γ:(3)$$\beta =\frac{\sum_{i=1}^{N_{j}}{\left|d_{j}\left(i\right)\right|}^{2}-N_{j}}{N_{j}},\;\gamma =\sqrt{\frac{1}{N_{j}}{\left(\frac{\ln N_{j}}{\ln2}\right)}^{3}}$$

If β is smaller than γ, the threshold function chosen is the sqtwolog threshold. Conversely, if β is greater than or equal to γ, the threshold function selected is the smaller value between the rigsure threshold and the sqtwolog threshold.

Threshold functions are typically classified into two types: hard threshold and soft threshold. The hard threshold function and soft threshold function are defined as Equation (4) and Equation (5) [35], respectively:(4)$${\hat{w}}_{j,k}=\left\{\begin{array}{c} w_{j,k},\left|w_{j,k}\right|\geq \lambda \\ 0,\;\;\;\;\left|w_{j,k}\right|<\lambda \end{array}\right.$$
(5)$${\hat{w}}_{j,k}=\left\{\begin{array}{c} \mathrm{sign}\left(w_{j,k}\right)\times \left(\left|w_{j,k}\right|-\lambda \right),\left|w_{j,k}\right|\geq \lambda \\ 0,\;\;\;\;\;\;\;\;\;\;\;\;\;\;\;\;\;\;\;\;\;\;\;\;\;\;\;\;\;\;\;\;\;\;\;\;\left|w_{j,k}\right|<\lambda \end{array}\right.$$

In both cases, coefficients with magnitudes smaller than the threshold are set to 0. However, the difference lies in how coefficients with magnitudes greater than the threshold are treated. In soft thresholding, coefficients with magnitudes greater than the threshold are contracted to 0 by subtracting the threshold value from the coefficient value. Conversely, in hard thresholding, coefficients with magnitudes greater than the threshold remain unchanged. Due to the fixed discrepancy between the wavelet coefficients processed by the soft threshold function and the original wavelet coefficients, there may be some distortion in the amplitude of the reconstructed signal. Hence, the hard threshold function is chosen.

For the selection of the wavelet basis function, the db3 wavelet is utilized. The db3 wavelet, developed by Ingrid Daubechies, a renowned scholar in wavelet analysis, is orthogonal, bi-orthogonal, tightly supported, approximately symmetric, and supports discrete wavelet transform. This wavelet serves as a sparse basis, introducing minimal smooth errors and ensuring a relatively smooth signal reconstruction process. In this study, a 3-level decomposition is employed. Figure 1 illustrates the wavelet decomposition process. The signal, denoted as S, is initially filtered using specialized low-pass and high-pass filters to generate a low-pass subband (A1) and a high-pass subband (D1). Following the Nyquist criterion, half of the samples are discarded. For subsequent decomposition levels, the same technique is iteratively applied to the low pass subband (A1) to produce the narrower subbands, A2 and D2, and so on.

The response signal from the FBG sensor was subjected to denoising using Matlab R2022a software. The impact of noise reduction in the wavelength time domain signal obtained from the impact is depicted in Figure 2.

## 3. Impact Monitoring System

### 3.1. Impact Monitoring Experimental Device

The experimental setup for impact monitoring consists of several components, including a portable computer for data storage, a carbon fiber composite tube with four FBG sensors attached to it, an optical sensing interrogator with a sampling frequency of 2000 Hz, and a drop hammer impact tester equipped with an anti-secondary impact device. A comprehensive depiction of the experimental setup is provided in Figure 3.

The real-time curve of the center wavelength of the FBG sensor was continuously monitored using the Fiber Grating Signal Processor software installed on the computer, and the necessary experimental data were recorded. The optical sensing interrogator utilized in the experiment has a detection wavelength range of 1528–1568 nm, a wavelength demodulation accuracy of ±1 pm, a wavelength demodulation resolution of 0.1 pm, and a sampling frequency of 2000 Hz. The working principle of the optical sensing interrogator is based on the reflection and interference effects of FBG sensors. Different wavelength waveforms generated by a scanning tunable laser are distributed to the corresponding sensing probes of FBG fiber optic gratings through a coupler. FBG sensors are highly sensitive to specific laser wavelengths corresponding to parameters such as temperature and pressure. The reflected signals are collected through the optical path, converted to electrical signals with optoelectronic conversion, and then processed by a demodulation system to reflect the parameters being tested by the system.

As shown in Figure 4, a scanning laser module can output multiple optical signals by utilizing a 1:8 optical splitter. Each optical signal can be connected to multiple FBG sensors with different center wavelengths. The scanning light source module operates in the C-band, scanning from 1528 nm to 1568 nm with a step size of 1 GHz or 8 pm in a periodic and rapid manner. The amplification and sampling circuitry, synchronized with the trigger output signal of the scanning tunable laser, can calculate the reflected wavelength of each individual FBG sensor in real-time, enabling demodulation.

All impact tests were conducted using a drop hammer impact tester model XL-300A, which is equipped with an anti-secondary impact device. A hemispherical punch with a diameter of 25 mm made of 45# steel was used in the tests. The total mass of the hammer head and hammer bar was 0.5 kg, and the impact energy was determined by the initial height difference between the punch and the test piece.

The experimental specimen used in the study was a resin-based carbon fiber composite tube with an inner diameter of 60 mm, an outer diameter of 66 mm, and a length of 300 mm. The tube was fabricated through roll molding, and the layup configuration was ${\left[0/90\right]}_{10}$. Prior to the experiment, the surface of the carbon fiber tube was cleaned with alcohol, and UV shadowless adhesive was used to affix the FBG sensors onto the tube’s surface. The condition of the FBG sensors was verified after the attachment process, and the four sensors were systematically placed around the experimental region. The positioning of the sensors is illustrated in Figure 5.

### 3.2. Acquisition of Impact Monitoring Signals

Considering the unavailability of a suitable fixture for the drop hammer impact tester, a custom test fixture was designed and fabricated specifically for the carbon fiber composite tube structure. The schematic diagram of the fixture clamping device is depicted in Figure 6. The carbon fiber composite tube specimen is securely held in place by rings and bolts at both ends of the fixture. During the low-velocity impact test, the specimen remains stable without any movement or overturning. Additionally, there is a defined gap between the specimen and the base of the fixture, ensuring that the specimen does not collide with the fixture’s base.

By sliding the fixture along the track, impacts can be applied at different points along the axial direction. To facilitate this, a grid is marked on the surface of the carbon fiber composite tube using a white paint pen. Considering the axisymmetric nature of the carbon fiber composite tube, a testing region with a circular center angle of 180° is selected. To accommodate the limitations of the drop hammer tester’s size, an experimental area with an 80 mm center width is chosen. The side surfaces of the carbon fiber composite tubes are unfolded to obtain rectangular regions, labeled as rows one to five from top to bottom and columns one to five from left to right, as illustrated in Figure 7.

The response signals from the FBG sensors are meticulously monitored and recorded throughout the experiment. A total of 21 impact test points are considered. Upon completion of the low-velocity impact test, a visual inspection is conducted near the impact location of the specimen to assess any damage on the impact surface.

### 3.3. Impact Test Results and Analysis

The center wavelength shift of the monitored FBG sensor exhibits variations corresponding to the applied impact load at different positions. As depicted in Figure 8, when impacting the (3, 3) position, all four FBG sensors observe a nearly identical small negative center wavelength offset. In Figure 9, at the impact (4, 3) position, FBG_A_ and FBG_B_ sensors record positive center wavelength offsets, while FBG_C_ and FBG_D_ sensors detect negative center wavelength offsets. Figure 10 demonstrates that at the impact (5, 3) position, FBG_A_ and FBG_B_ sensors show minimal center wavelength changes, while FBG_C_ and FBG_D_ sensors observe significant positive center wavelength offsets. Similarly, in Figure 11, at the impact (5, 4) position, FBG_A_ sensor records a small positive center wavelength shift, FBG_B_ sensor exhibits a minimal center wavelength shift, FBG_C_ sensor detects a positive center wavelength shift, and FBG_D_ sensor observes a large positive center wavelength shift.

## 4. Low Velocity Impact Monitoring Based on BP Neural Network

### 4.1. Overview of BP Neural Networks

BP neural networks possess strong nonlinear mapping capabilities, high self-learning and self-adaptive capabilities, the ability to apply learning to new knowledge, and a certain degree of fault tolerance. These networks are capable of learning and storing numerous input–output pattern mapping relationships without prior knowledge of the mathematical equations describing such relationships. The learning rule of the BP network involves continuously adjusting the network’s weights and thresholds through backpropagation using the steepest descent method to minimize the sum of squares error. Its distinctive feature lies in the forward propagation of signals and the backward propagation of errors.

The BP network consists of three layers: an input layer, a hidden layer, and an output layer. The input layer receives input data, the hidden layer processes information (with the option to include multiple layers), and the output layer provides the final result. Figure 12 illustrates the flowchart of the BP neural network algorithm.

The process of the BP neural network can be divided into two phases. The first phase involves the forward propagation of signals, starting from the input layer, passing through the hidden layer, and finally reaching the output layer. The second phase is the reverse propagation of errors, starting from the output layer, passing through the hidden layer, and finally reaching the input layer. In this stage, the weights and biases from the output layer to the hidden layer, as well as from the hidden layer to the input layer, are adjusted sequentially.

### 4.2. Results and Analysis

A BP neural network was developed using MATLAB software to predict the impact location of a carbon fiber composite tube. The network was trained using a total of three datasets, each consisting of 21 data points, resulting in a total of 63 sets of data. The maximum observed center wavelength shift was used as the input, while the axial (X) and radial diffusion coordinates (Y) of the carbon fiber tube were used as the output. The network architecture consisted of a single hidden layer with five nodes. During the training process, the maximum iteration value was set to 1000, the minimum training target error was 1e−6, and the learning rate was set to 0.01.

The BP neural network was trained using the experimental dataset, and its performance was evaluated by randomly selecting four cases for testing. The predicted impact locations, considering the axial direction of the carbon fiber composite tube as the horizontal coordinate and the unfolded length in the diameter direction as the vertical coordinate, are presented in Table 1. The maximum positioning error was found to be 22.82 mm, while the average positioning error was 20.68 mm.

## 5. Low Velocity Impact Monitoring Based on Error Outliers

### 5.1. Overview of Error Outlier Algorithms

Based on the response signals obtained from the low-velocity impact test, it is evident that there is a significant disparity between the response signals acquired at different impact positions, while the difference between the response signals acquired at the same impact position is minimal. Figure 13 illustrates a comparison of the errors between signals at the same impact position and signals at different impact positions. It is evident that the peak error for signals from different impact positions is larger compared to the peak error calculated between signals from the same impact position.

To determine the impact location, the error is calculated between the reference signal and the random test signal. Then, outliers are identified by comparing the error with a predefined threshold. Finally, the impact location is determined based on the outliers. The flowchart of the impact localization algorithm based on error outliers is presented in Figure 14.

Initially, the acquired impact response signal undergoes preprocessing. The preprocessed response signal is standardized using Equation (6), where S represents the response signal, S_mean_ represents the mean value of the response signal, and S_std_ represents the standard deviation of the response signal. The error between the standardized reference signal and the random test signal is calculated using Equation (7):S_s_ = (S − S_mean_)/S_std_(6)
E_(*n*,i)_ = S_Ref.*(n*,i)_ − S_Test(i)_(7)

The initial point index *n* = 1 of the response signal is changed to *n* = 100 with an increment of Δ = 1. At each moment, the error between the reference signal S_Ref.(n,i)_ and the random test signal S_Test(i)_ is calculated point by point. The magnitude of the calculated error is compared to a predefined threshold to determine the magnitude of outliers at the 21 points in the grid. This process is repeated for the response signals acquired by the four FBG sensors used in the test. The outlier size for each location is determined by summing the outliers for each location, monitored by all FBG sensors. To determine the location with the least outliers, a minimum of three locations is required. The predicted locations are calculated using Equations (8) and (9), where n represents the number of selected positions, $O_{i}$ represents the size of the outlier at the corresponding position, $x_{O_{i}}$ represents the x-coordinate of the corresponding position, and $y_{O_{i}}$ represents the y-coordinate of the corresponding position, as follows:(8)$$P\left(O_{i}\right)=\frac{1-\frac{O_{i}}{\sum_{i=1}^{n}O_{i}}}{n-1}$$
(9)$$x=\sum_{i=1}^{n}P(O_{i})\times x_{O_{i}},y=\sum_{i=1}^{n}P(O_{i})\times y_{O_{i}}$$

### 5.2. Selection of the Error Threshold

The number of outliers obtained at each reference location using different error thresholds is presented in Figure 15. It is evident that the number of outliers is highly sensitive to the selected error threshold. Lower thresholds result in more outliers, while higher thresholds lead to fewer outliers.

To determine the appropriate error threshold for the outlier-based impact localization algorithm, the impact localization performance was tested using error thresholds ranging from one to six. The summary of the variance of the positioning error and the localization error is presented in Table 2. The localization error does not differ significantly when the error thresholds are set to one, two, and five. However, the variance is minimized, and the localization stability is improved when the error threshold is set to five. Consequently, an error threshold of five is adopted for the impact localization algorithm.

### 5.3. Results and Analysis

Overall, four random cases were chosen for model testing. The total sum of error outliers for FBG_A–D_ was visualized using a color map, illustrating the number of outliers at each reference point in the specimen, as shown in Figure 16. The color map ranges from dark blue to dark red, with dark blue indicating a small number of outliers and dark red indicating a large number of outliers. Locations closer to the actual impact location are represented by colors closer to dark blue.

The impact localization results obtained using the error outlier-based algorithm are presented in Table 3. The average localization error was found to be 13.98 mm, and the maximum localization error was 14.70 mm.

## 6. Conclusions

This research investigates the utilization of FBG sensors for monitoring low-velocity impacts on carbon fiber composite tubes. When a low-velocity impact occurs, the center wavelength of the FBG sensor shifts and is captured by an optical sensing interrogator. The response signal from the low-velocity impact is denoised using the discrete wavelet transform. The variations in the response signals captured by the FBG sensors are analyzed for their different impact locations.

The low-velocity impact monitoring method based on the BP neural network utilized the maximum offset of the monitored center wavelength as the input for neural network training. The impact coordinates were used as the output for training, and a total of three data sets were employed to train the neural network. In total, four cases were randomly selected for testing, and the results revealed an average positioning error of 20.68 mm for determining the impact location of the carbon fiber tube.

The low-velocity impact monitoring method based on error outliers involves calculating the error between the impact signal and a reference signal (using only one data set) point by point at each time moment. This error is then compared to a predefined threshold to determine the number of outliers. A smaller number of outliers at a specific reference point indicates proximity to the impact location. In total, four randomly selected cases were tested using this method, resulting in an average localization error of 13.98 mm.

A comparison is made between the BP neural network-based and error outlier-based algorithms for low-velocity impact localization. The latter demonstrates higher prediction accuracy and does not require a large amount of data.

## Figures and Tables

**Figure 1 sensors-24-01279-f001:**
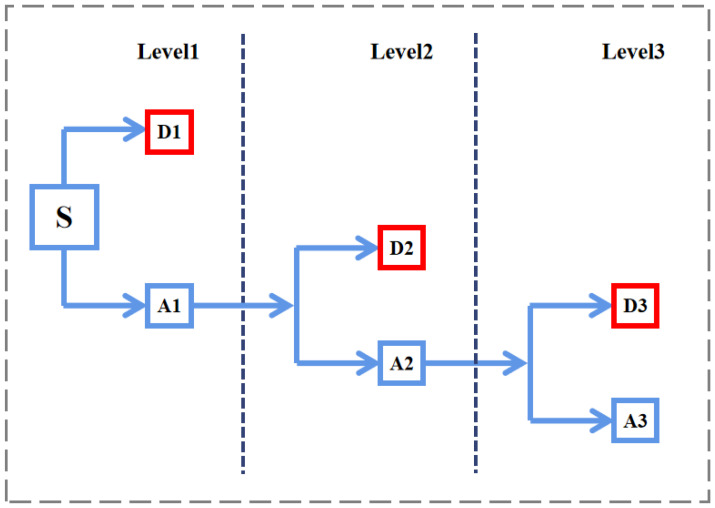
Wavelet decomposition at a scale of three.

**Figure 2 sensors-24-01279-f002:**
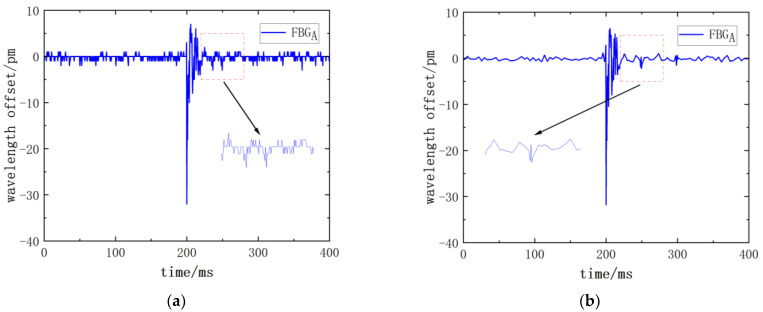
The influence of discrete wavelet transforms denoising on the response signal of FBG sensor (**a**) before signal denoising and (**b**) after signal denoising.

**Figure 3 sensors-24-01279-f003:**
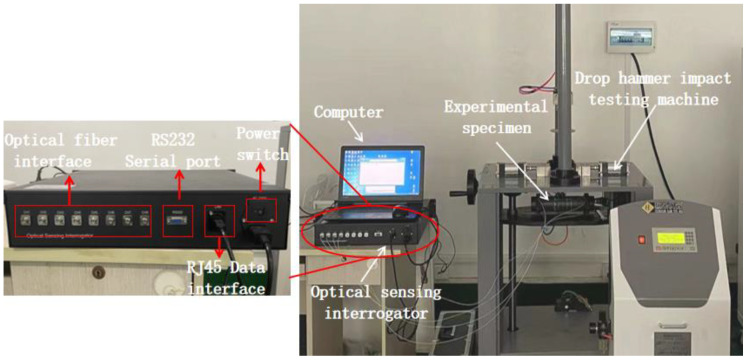
Experimental setup.

**Figure 4 sensors-24-01279-f004:**
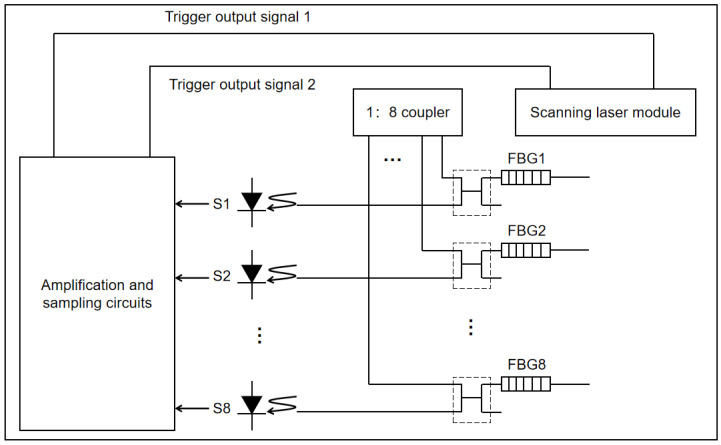
Schematic diagram of the application principle of optical sensing interrogator.

**Figure 5 sensors-24-01279-f005:**
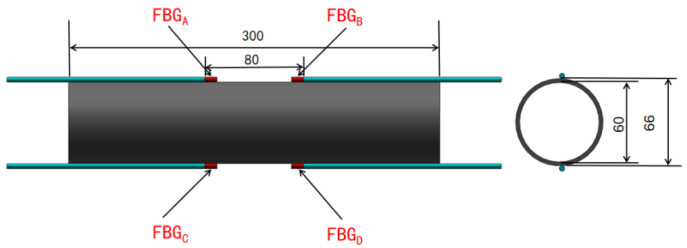
Schematic diagram of FBG sensor arrangement.

**Figure 6 sensors-24-01279-f006:**
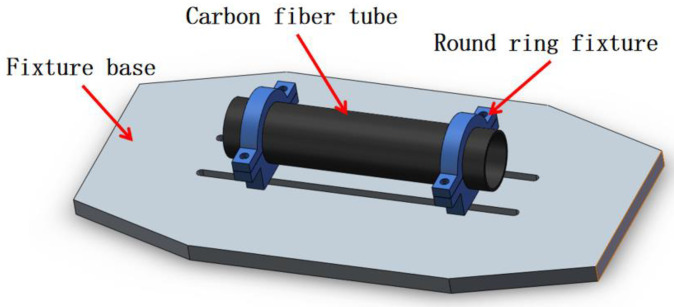
Schematic diagram of fixture clamping.

**Figure 7 sensors-24-01279-f007:**
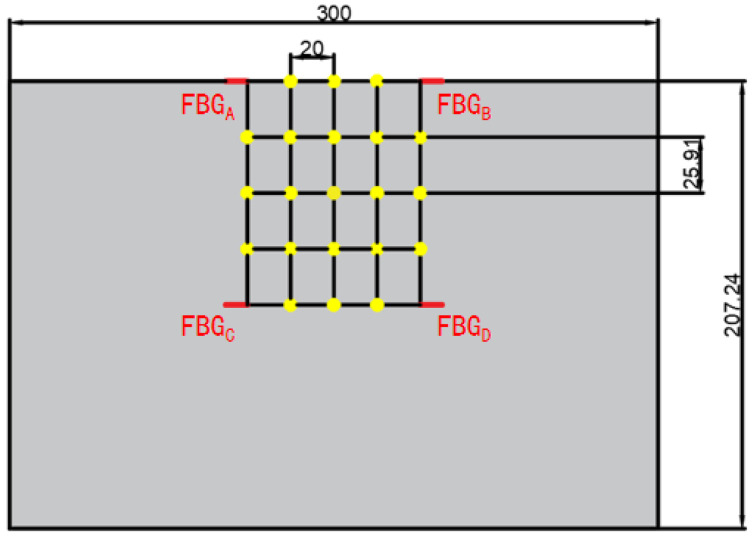
Schematic diagram of the layout of sensors in the impact region and the arrangement of impact experiment points.

**Figure 8 sensors-24-01279-f008:**
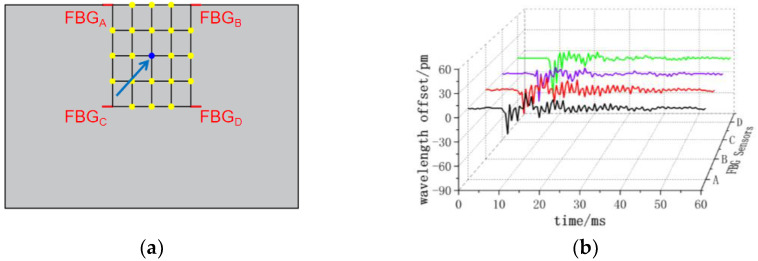
(3, 3) position impact center wavelength offset value (**a**) impact position; (**b**) FBG sensor center wavelength offset value.

**Figure 9 sensors-24-01279-f009:**
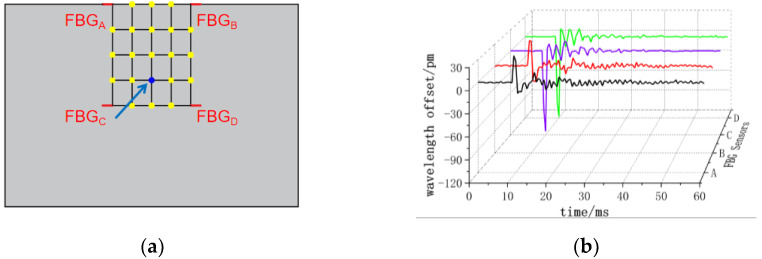
(4, 3) position impact center wavelength offset value (**a**) impact position; (**b**) FBG sensor center wavelength offset value.

**Figure 10 sensors-24-01279-f010:**
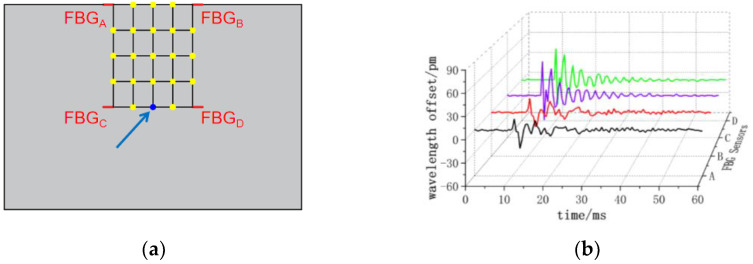
(5, 3) position impact center wavelength offset value (**a**) impact position; (**b**) FBG sensor center wavelength offset value.

**Figure 11 sensors-24-01279-f011:**
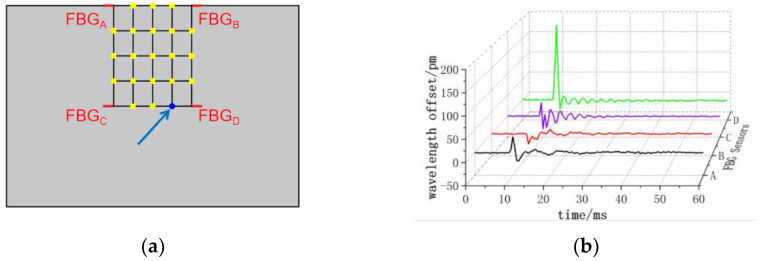
(5, 4) position impact center wavelength offset value (**a**) impact position; (**b**) FBG sensor center wavelength offset value.

**Figure 12 sensors-24-01279-f012:**
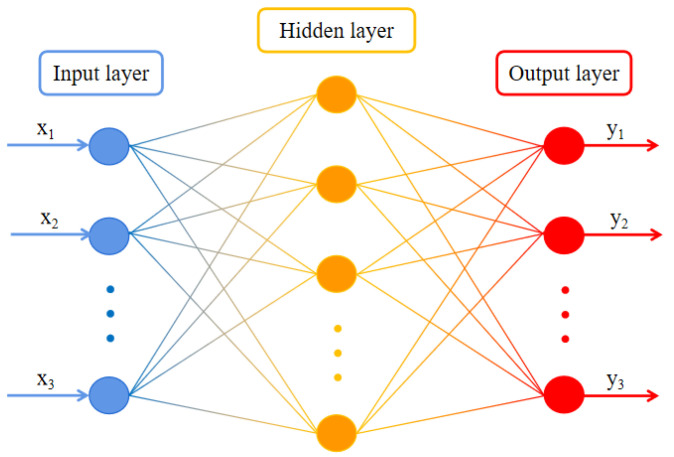
Flowchart of BP neural network algorithm.

**Figure 13 sensors-24-01279-f013:**
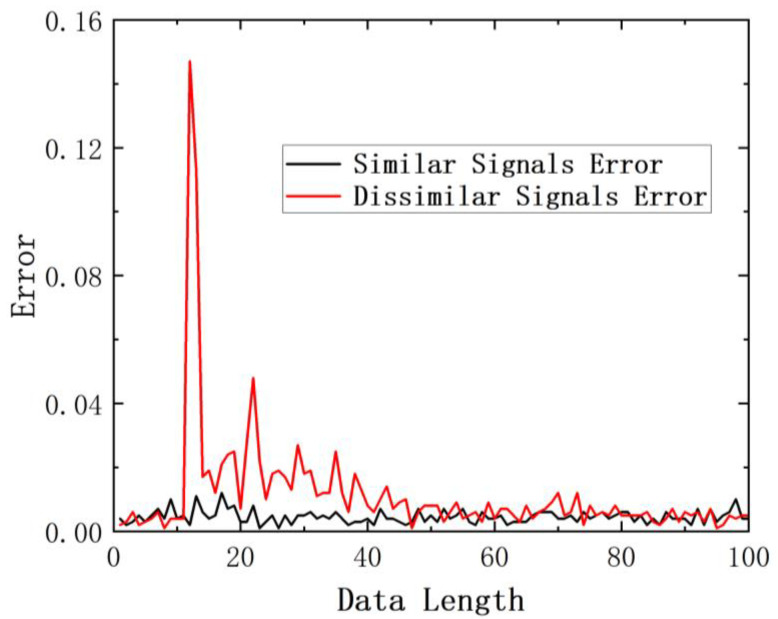
Comparison of errors between signals from the same impact position and different impact positions.

**Figure 14 sensors-24-01279-f014:**
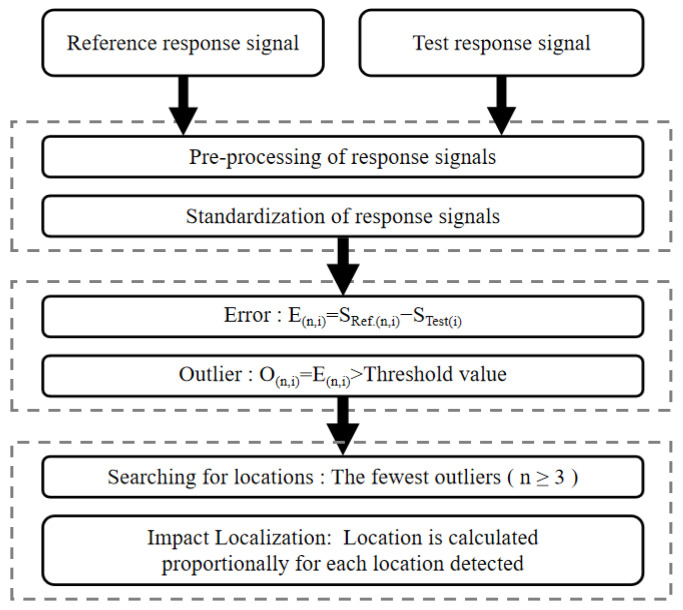
Flowchart of outlier based localization algorithm.

**Figure 15 sensors-24-01279-f015:**
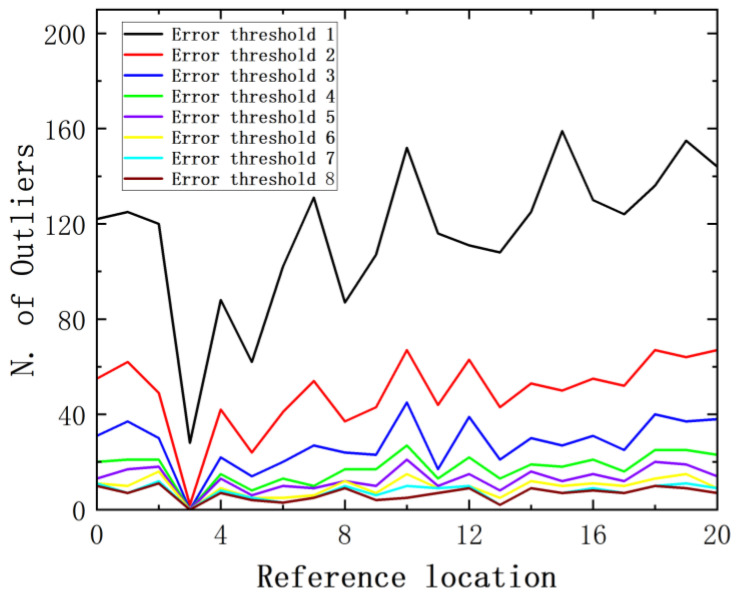
Outliers when impacting different reference positions.

**Figure 16 sensors-24-01279-f016:**
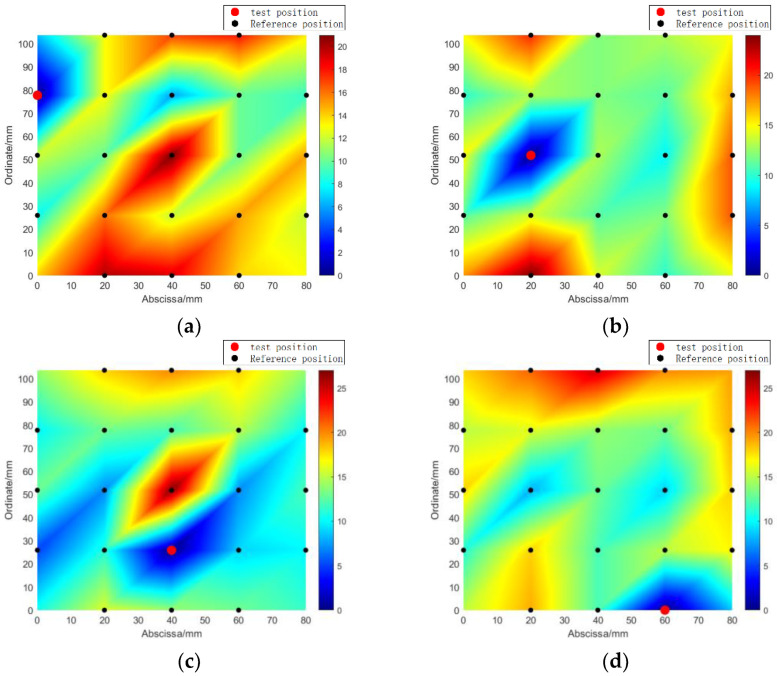
The number of outliers corresponding to different impact locations (**a**) impact locations (2, 1); (**b**) impact position (3, 2); (**c**) impact position (3, 3); (**d**) impact position (5, 4).

**Table 1 sensors-24-01279-t001:** Impact localization results based on BP neural networks.

Impact Location (mm)	Predicted Location (mm)	Localization Errors (mm)
(20, 77.715)	(34.74, 67.58)	17.88
(40, 77.715)	(29.61, 57.40)	22.82
(40, 103.62)	(39.13, 83.16)	20.48
(60, 0)	(56.09, 21.17)	21.53
Average error		20.68

**Table 2 sensors-24-01279-t002:** Impact localization results at different error thresholds.

Error Threshold	Average Error (mm)	Variance
1	13.76	21.33
2	13.11	65.63
3	15.63	11.70
4	18.35	42.22
5	13.98	1.17
6	14.88	105.13

**Table 3 sensors-24-01279-t003:** Impact localization results based on error outliers.

Impact Location (mm)	Predicted Location (mm)	Localization Errors (mm)
(10, 77.715)	(10.40, 83.93)	12.12
(20, 51.81)	(33.56, 46.15)	14.70
(40, 25.905)	(30.16, 36.81)	14.68
(60, 0)	(54.55, 13.35)	14.42
Average error		13.98

## Data Availability

The raw/processed data required to reproduce these findings cannot be shared at this time as the data also forms part of an ongoing study.

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
