# Peer review of "Low Velocity Impact Monitoring of Composite Tubes Based on FBG Sensors"

_sensors, 2024, doi:10.3390/s24041279_

Round 1
Reviewer 1 Report
Comments and Suggestions for Authors
In this work, authors detected impact signal through fiber Bragg grating (FBG) affixed to the surface of a carbon fiber composite tube. The hidden damage detection from low velocity external impacts of Carbon fiber reinforced composites (CFRP) can be realized through capturing the central wavelength shift. In addition, error outlier algorithm based method is proposed to decrease noise. Compared to BP neural network method, there is higher prediction accuracy of error outlier algorithm based method in average localization error. The description of work in experiment is integrity. However, there is lack of full expression of the description of necessity and that of logic. In conclusion, I recommended major revision of this manuscript. The specific problems are as follows:
1. In the part of introduction, authors should pay more attention to the accuracy of damage detection instead of detection method. In addition, authors should conclude the problems of former research in order to emphasize the importance of the manuscript. I suggest authors re-write the introduction part.
2. There are some format problems:
(1) Figures and their legends should be at the same page.
(2) In figure 7-10, the specific impact value should be shown in the manuscript. In addition, the national standard should be figured out.
3. A few works regarding to the sensing using the nanostructures are suggested to reference,
“A Reversible Tuning of High Absorption in Chalcogenide–Metal Stacked-Layer Structure and Its Application for Multichannel Biosensing”. Advanced Photonics Research, 2021, 2(8): 2000152.
“Mid-infrared supercontinuum generation in chalcogenide glass fibers: a brief review.” PhotoniX 2, 9 (2021). https://doi.org/10.1186/s43074-021-00031-3
Comments on the Quality of English Language
Author Response
"Please see the attachment."

Reviewer 2 Report
Comments and Suggestions for Authors
This manuscript introduces a method that utilizes fiber Bragg grating sensors and neural network/error outliers for monitoring low-velocity impacts and determining the impact locations on carbon fiber composite tubes. The article is well-structured and organized. However, I have several inquiries that need clarification from the authors:
1. The section index should start from 1 rather than 0.
2. Other optical fiber sensing techniques, such as optical frequency domain reflectometry (OFDR), can perform fully distributed strain measurements using a single strand of optical fiber with high spatial resolution. This technique can precisely predict impact locations. Please add some references, compare the advantages of different techniques, and clarify the reasons for choosing FBGs in this study (DOI: 10.1109/JSEN.2022.3197730).
3. Why do you use neural network/error outliers to monitor the impact locations instead of relying solely on optical fiber sensors? Analyzing the strain relationships among the four FBGs to determine the impact locations, as described in Lines 217-227, seems feasible. The rationale behind the monitoring method used in this study should be clarified.
4. The model of the FBG interrogator should be specified. A schematic of the interrogator, including the laser source, multiple measurement channels, photodetectors, and DAQ, etc., should be provided alongside Figure 3.
5. The material of the hammer head/punch should be specified, as different materials may lead to distinct impact characteristics, especially in distributed acoustic sensing.
Author Response
"Please see the attachment."

Reviewer 3 Report
Comments and Suggestions for Authors
This article reports on the impact monitoring of composite pipes based on the principle of fiber Bragg grating reflection. BP neural network and error outlier algorithm are used to predict the location of low velocity impacts. The results are well presented, however, some points should be addressed before considering its publication.
1. In the acquisition of impact monitoring signals, four FBGs are used. In Fig. 6, thet illustrate the locations of the 4 FBGs, and the 21 impact test points. It is not clear how the impact points affect the FBG differently. Why some FBG exhibits negative center wavelength offset, while the others exhibits positive shift.
2. The localization errors of these two methods are 20.68 mm and 13.98 mm. How are these errors compared with other reported works? A table making the comparison might help authors to see the improvement made in this work.
3.The waveforms in Figures (2), (7) - (10), (12), and (14) are too small to be easily observed. The emphasized or expressed parts in the diagram should be enlarged and highlighted, and their coordinate axes should be clear
Author Response
"Please see the attachment."

Reviewer 4 Report
Comments and Suggestions for Authors

Author Response
"Please see the attachment."

Reviewer 5 Report
Comments and Suggestions for Authors
The authors submitted a paper describing research into carbon fiber reinforced composite materials using fiber Bragg gratings. My comments regarding this work are as follows:
1. I would suggest that authors build semantic connections within the abstract so that it is read better. For example, it first says that a well-known type of pointwise sensor is being used, then it immediately says that error values have been obtained for determining the length - as if a distributed sensor were used. If you do not refer to the full text of the article, then such an abstract looks incomplete.
2. Introduction is point number 0. Is this possible?
3. In paragraph 1.2. the authors 'jump' right into discussing DWT. Of course, this type of noise reduction is now quite common in science and technology. But it is necessary to explain why the data cannot be processed by other methods? Why is convolution with a basis (Wavelet) function important, and not with a periodic one, as in the Fourier transform? In my opinion, it would be appropriate to expand the paragraph where noise reduction is discussed and consider, for example, moving average (MA), moving differential (MD), frequency domain dynamic averaging (FDDA) and/or activation function dynamic averaging (ADFA), which work excellent not only in the frequency domain, but also in the time domain.
4. I would ask the authors to turn the inscription in Figure 3 into white color, since the text is very difficult to read.
5. It is necessary to know the design or model of interrogator since the experiment must be repeatable. I ask the authors to write more about the optical part of the installation.
6. With what accuracy was the localization standard (reference) determined? Is it possible to use optical frequency domain reflectometry (OFDR) for this? If this is possible, then the reference localization would be accurate to tens of microns.
7. In almost all graphs, a very small font is chosen for labeling the axes. Of course, the article is an electronic document that can be enlarged at any time, but this is not always convenient when reading the text.
8. Graphs and tables in the text are located mainly at the end of sections; I suggest raising them higher, immediately after the first mention.
9. Please check the text for the typos, missing spaces, etc...
Author Response
"Please see the attachment."

Round 2
Reviewer 1 Report
Comments and Suggestions for Authors
The authors have properly addressed my concerns and I suggested it to be accepted.
Comments on the Quality of English Language
I think the grammar needs to be further polished.
Reviewer 2 Report
Comments and Suggestions for Authors
All the comments and questions have been addressed properly.